# Sword and Lotus: The Life of a Confucian Buddhist Woman Warrior in Seventeenth Century China

**Hongyu Wu**

School for the Humanities and Global Cultures, Ohio Northern University, Ada, OH 45810, USA; h-wu@onu.edu

**Abstract:** This paper focuses on the life of Liu Shu (1520–1657), a woman warrior who lived through the social and political turmoil of the violent dynastic transition from the Ming (1368–1644) dynasty to the Qing (1636–1911) dynasty. Drawing on the writings of Liu Shu and on different versions of accounts of her life written by male literati in different time periods, this paper intends to reveal the multiplicity and complexity of how a woman could exert her agency and interact with the dominant structure. Commonly, women's agency has been understood as resistance against the male-dominant patriarchal system. However, recently, scholars such as Saba Mahmood have problematized universalizing overtly resistant acts against the patriarchal society to bring radical changes as demonstrations of women's agency. These scholars argue that this approach fails to recognize that through their autonomy and while living a life of self-fulfillment, women have the capacity to reproduce, sustain, or subtly change the social norms and views of values that justify and support the patriarchal structure. In light of these scholars' studies, this paper explores Liu Shu's engagement in political and military activities and her Buddhist practices to analyze how she transgressed established gender norms in order to uphold rather than reject the virtues promoted by patriarchal ideology. The paper also discusses how conflicting demands on women by a male-centered society in the drastic dynastic transition enabled her to negotiate with or challenge the dominant structure. It also considers how her disobedient acts were accepted and applauded by some male literati to address their own agenda in different cultural, historical, and political contexts.

**Keywords:** Confucian gender norms; agency; filial piety; chastity; loyalty; Confucianism; Buddhism





## 1. Introduction

The violent dynastic transition from Ming (1368–1644) to Qing (1644–1911) brought about loss of life, dislocation, and other kinds of traumatic experiences. However, the chaos also provided some possibilities to women that were unavailable in a stable society, for example, crossing gender boundaries through their physical or literary work (Li 2010, p. 179). The life of Liu Shu (劉淑 1620–1657), a woman poet and warrior who lived during the turmoil of the Ming–Qing transition provides a good example for demonstrating how a woman could push gender boundaries, venture into the normally male-defined spheres of her time—such as warfare and politics—and play roles generally assigned to men during this period. This paper focuses on representations of Liu Shu's political engagement and religious life that appear in her own writing and that of male literati. It is almost impossible to accurately reconstruct her life through these writings, but the representations of her activities in these writings reveal multiple ways that women could exert their agency to live an independent and empowered life in a patriarchal society.

Commonly, women's agency has been understood as resistance against the male-dominant patriarchal system. Only in recent years have scholars such as Saba Mahmood problematized the universalization of overtly resistant acts against the patriarchal system to bring about radical changes as the demonstrations of women's agency. According to Mahmood, this resistance model of agency fails to recognize that women can live a life of

self-fulfillment through reproducing, sustaining, or subtly changing the social norms and views of values that justify and support the patriarchal structure (Mahmood 2005, pp. 5–10). Other than intentional resistance against it, Liu Shu's engagement in political, military, and religious activities and the portrayal of these activities in her writing and that of male literati reveal a more complicated way that a woman could interact with the patriarchal system. Nevertheless, patriarchal institutions are not always consistent in their requirements for women's virtues—especially in a time of dynastic transition and associated shifts of political allegiance. As a consequence, women can have more space to negotiate gender boundaries and redefine gender norms and gender roles. This is evidenced in Liu Shu's life and her literary works.

In recent years, increasing numbers of scholarly works have investigated how women in imperial China pushed gender boundaries through their engagement in political and social activities. Dorothy Ko's research shows women's engagement in cultural and social activities through their literary works and social networking that transcended the domestic sphere (Ko 1995). Grace Fong (Fong 2008), Susan Mann (Mann 1997), and Maram Epstein (Epstein 2011) also demonstrate how women authors deployed their writings to negotiate gender boundaries, challenge gender norms in a subtle way, or redefine femininity and masculinity. In addition to the study on women authors, scholarly attention also extends to another type of women who were also actively involved in public affairs not through their brushes but through their swords and lances—the women warriors. Shoufu Yin's study on Princess Pingyang's involvement in warfare (Yin 2021) of the Tang dynasty, Pei-yi Wu's study on Yang Miaozhen, a female military commander of the Song–Yuan transition (Wu 2002), and Louise Edwards's research on the legendary woman warrior Mulan (Edwards 2010) all show that the patriarchal society tolerated, accepted, and extolled these women warriors by manipulating their narratives in keeping with various motivations; however, these scholars based their research on official documents, biographies, dramas, and poems written by men. As such, the voices of these militant women are totally absent in their studies. The fact that most of these women military commanders did not write is an obvious reason for the absence of women's voices. In contrast, Liu Shu provides a good case that bridges the gap between women authors and women warriors. Through her study of Liu Shu's poems, Wai-yee Li illustrates how political disorder created a space in which women could create new forms of expression and attain opportunities to step outside of the designated gendered sphere to fulfil their heroic aspirations (Li 2010, p. 179). In the light of the work of these scholars, this paper aims to add another dimension for looking at the life and experience of the woman warrior. In particular, it considers Liu Shu as a Confucian Buddhist woman warrior: how did she wield Confucian virtues to justify her transgressive behavior and how did she creatively employ Buddhist concepts and practices to express her political allegiance and transcend the turmoil and pain experienced in the disorder brought about by dynastic transition?

## 2. Introduction: The Life Account of Liu Shu

In addition to Liu Shu's self-expressive literary works, her life is known to us through biographies composed (or compiled) by male literati in different time periods. Male literati, such as Zou Yi 鄒漪 (sixteenth century) in his *Qizhen ye sheng* 啓禎野乘 (Unofficial History of the Tianqi and Chongzhen Era, first printed in 1666, then reprinted in 1679), Chen Weisong 陳維崧 (1626–1682) in his *Fu ren ji* 婦人集 (Writing Women), Peng Shaosheng 彭紹升 (1740–1796) in his *Shan nüren zhuan* 善女人傳 (Biographies of Good Women), and Li Yao 李瑤 (ca. Daoguang reign 1820–1850) in his *Nanjiang yi shi* 南疆繹史 (the History of the Southern Region), all had an entry of Liu Shu's life in their works. Except for one or two differences, the versions by Zou Yi, Peng Shaosheng, and Li Yao are similar. Chen Weisong's version was shorter and did not mention her life after her failed military action, while Zou Yi, Peng Shaosheng, and Li Yao wrote about Liu Shu's Buddhist life after her military engagement in a very brief way. It is hard to tell which version is more histori-

cally accurate, but examining how these male writers represented Liu Shu's life helps us understand the intellectual/ideological agenda of the male literati who detailed her deeds.

In this paper, I choose to use Peng Shaosheng's version as an example that is more relevant to my discussion of Liu Shu's Buddhist life. I will start with Peng Shaosheng's biography of Liu Shu collected in his *Shan nüren zhuan*, a biographical collection of exemplary Buddhist laywomen. He also acknowledged that his account of Liu Shu's life was based on Zou Yi's biography. To compare Peng's biographical account of Liu Shu and her own writings enables us to see how Liu Shu led a life of autonomy through upholding values that reinforced the Confucian patriarchal system.

Peng Shaosheng described Liu Shu's life in the following way:

Liu Shu 劉淑 [1620–1657] was the daughter of the famous Ming martyr Liu Duo 劉鐸 [1573–1626] of Luling 盧陵 [in the Ming dynasty]. On the night she was born, the room filled with a wonderful fragrant smell, lasting the whole night. She could sit gracefully when a few months old. She did not cry or laugh casually like other young children. At the age of four or five, she could read such Confucian classics as the *Xiao jing* 孝經 [Classic of Filial Piety] and *Lun yu* 論語 [*Analects*]. When Liu Shu was seven, her father was the magistrate of Yangzhou 揚州 and was falsely charged by the eunuchs. When her father was arrested and taken to Beijing, the capital, Liu Shu begged her mother to let her follow her father. As soon as they arrived in Beijing, Liu Shu wrote an appeal for her father in her own blood. She planned to beat the drum outside of the imperial court to submit the appeal to the emperor but was stopped by her mother. Upon receiving the news that her father had been executed, Liu Shu attempted to kill herself. Fortunately, her nanny promptly stopped her. After she went back to her hometown and buried her father, she read all the books he had left behind. She favored Buddhist scriptures most. Her interests extended to books on swordsmanship and military strategy by Sun [Wu] and Wu [Qi]]. A few years later, she married Wang Cixie 王次偕 of the same county and gave birth to a son. Cixie died young. In 1644, when [the rebels led by] Zhang Xianzhong [1606–1647] captured Hunan, [neighboring] Jiangxi [where Liu Shu lived] was jeopardized and people were on alert. In the same year, the capital was taken over by another army of rebels led by Li Zicheng [1606–1645]. With funds obtained from selling all her property, Liu Shu organized an army of several hundred soldiers, made up of fighters she enlisted and the guards of her family, to fight against the rebels and seize back the capital city. She organized her army according to the well-known *Sima Military Strategy* 司馬法. Before she led her army across the river to the north, she considered recruiting more allies. However, most of the local army officers hesitated to join her. A General Zhang from Yunnan happened to arrive at Jiangxi at that moment. Liu Shu went to see him to discuss their cooperation to fight against the rebels and seize back the capital. She also generously rewarded General Zhang's soldiers to mobilize them to fight against the rebels. However, the general was not so determined to fight. As General Zhang was of two minds, disputes erupted between her and the general. Angered by the general, Liu Shu pulled out her sword to stab the general. General Zhang's soldiers were alerted and put on their arms. With the mediation of her aides, [a fight between the two sides did not erupt]. Eventually Liu Shu had to give up her plan. She dispersed her own soldiers and withdrew to her hometown, where she built a small hut called "Lotus Ferry". She donated a large piece of land to a Buddhist temple and commissioned monks to hold memorial services for the deceased. She adopted a vegetarian diet and chanted Buddhist sūtras in her small hut for the rest of her life. (Peng 1873, vol. 2, pp. 22b–23b)

Peng Shaosheng portrays Liu Shu as a filial daughter as suggested by activities such as her writing a petition in her own blood on behalf of her father and then attempting to kill herself for her father's sake. What is equally obvious in this story is that she was

a woman warrior and a loyal subject who courageously fought to save the country from rebels (and the Manchus who occupied the capital three month after the rebels took over the city). However, her life as a Buddhist laywoman is not the center of Peng's narrative. Both Zou Yi and Li Yao's versions also include these activities. In this sense, it is safe to assume that the three male biographers intended to depict Liu Shu as a filial daughter and a loyal subject. Interestingly, according to Liu Shu's writing, it was her mother who wrote an appeal in blood on behalf of Liu Duo, and it was also her mother who planned to kill herself to follow Liu Duo in death (Liu 1992, pp. 363, 365). Why, then, did Peng Shaosheng attribute these behaviors to Liu Shu?

Peng Shaosheng (dharma name: Jiqing 際清) was a Confucian-turned Buddhist layman living in the eighteenth century, a time period during which the intellectual, social, and political milieu was not always favorable to Buddhism. In the intellectual field, debates between Confucians and Buddhists had been going on for centuries. Peng himself was involved in written debates with prominent Confucian scholars and literati such as Dai Zhen 戴震 (1724–1777) and Yuan Mei 袁枚 (1716–1797). Dai Zhen was a leading figure of the *Kao zhen xue pai* 考證學派 (Evidential School), a very influential school in the southern Yangzi River area where Peng spent most of his life. This school promoted the recovery of the authentic meaning of the Confucian classics through philological study. Moreover, it only accepted interpretations of the classics that predated the introduction of Buddhism into China. For Dai Zhen, the Confucian thinkers of the Song and Ming dynasties were so heavily influenced by Buddhism that their interpretations of Confucian classics were laced with terminology tainted by Buddhism. In Dai Zhen's view, this not only distorted the meaning of the classics; it also tended to substitute personal opinions for universal moral truth. Thereby, it brought great harm to social morality (Dai 1980, p.173).

In Peng's day, Yuan Mei was a famous writer and literary critic; he was also a family friend. In his correspondence with Peng, Yuan accused Buddhism of selfishness and ignoring one's responsibilities for one's rulers and family (Yuan 1993, p. 340). Some members of the political elite—such as Tang Bin, who was a colleague of Peng Shaosheng's great-grandfather—also accused Buddhism of leading to violation of the gender segregation prescribed by Confucian behavior codes. In their view, such violations resulted in moral degeneration and the undermining of the stability of family and society (Tang 1991, p. 31).

To defend Buddhism, Peng Shaosheng argued Buddhism was in complete harmony with Confucianism. Throughout his writings, Peng insists that attainment of the Buddhist spiritual goal—that is, enlightenment or rebirth in the Pure Land—lay in the fulfillment of secular responsibilities as a family member and as a subject of the realm. He claims that Buddhist practices, in fact, enable people to fulfil these duties better (Peng 1983, p. 151b12–15). The biographical collection of *Shan nüren zhuan* is part of the effort to defend Buddhism and argue for the compatibility of Buddhism and Confucianism. The story of Liu Shu furnishes a good example of Peng's agenda.

As previously mentioned, Peng's biography of Liu Shu was adapted from Zou Yi's account of Liu Shu in his *Qi zhen ye cheng*. Zou's book was indeed a non-Buddhist collection of biographical narratives about upright officials, famous literati, women martyrs, and chaste widows from the Tianqi era (1621–1628) to the Chongzhen era (1628–1644) of the Ming dynasty. Most of Peng's account is the same as the original text written by Zou Yi, except for a few differences.

In Peng's version, he omits details about the conflict between Liu Shu and General Zhang, and it seems that the dispute was caused by the reluctance of the general to fight for the country. Zou Yi's version gives a more detailed description of the conflict. It says, "The general spoke to her disrespectfully 語不遜", (Zou 1936, vol. 2, p. 16a) implying a possible sexual advance to which Liu Shu responded with her fury and a sword. However, Liu Shu herself describes the incident vaguely: "I did not expect the treacherous official to display his evil intentions. I vowed to die. However, considering my father's unburied remains and my mother's loving kindness, [I gave up my will to die]. Blessed by the spirit of my father, I was able to escape from the danger" (Liu 1992, p. 366).

Peng's version places greater emphasis on Liu Shu's filial devotion to her father and her loyalty to the Ming emperor but downplays the conflict. His omission of the alleged sexual advance is probably related to the accusation of the violation of gender segregation that Confucian officials made against Buddhists. The purpose of prohibiting the intermingling of men and women was to defend women against unwanted male sexual advances or to prevent licentious relationships between men and women. Thus, Liu Shu's venturing into the male domain would have been viewed as trespassing gender segregation with the potential of jeopardizing her chastity. In this sense, it would not have been appropriate for a biography of an exemplary Buddhist laywoman. For Liu Shu, this incident was humiliating and traumatic. Not surprisingly, she did not write about this incident in detail. In contrast to Peng's version, Zou Yi highlighted the disputes and Liu Shu's heroic resistance against the general's sexual advance to represent her not only as a filial daughter and a loyal subject, but also as a heroic woman who furiously defended her chastity. Living in the time of the dynastic transition from Ming to Qing as Liu Shu was, Zou Yi highly praised Liu Shu's courage and determination to defend her chastity, the feminine virtue that was equivalent to the loyalty of male subjects to their ruler.

Another revision made by Peng Shaosheng also deserves our attention. Zou Yi writes, "[Liu Shu] read all the books her father left behind day and night. Her interest extended to Chan Buddhism, swordsmanship and military strategy books by Sun [Wu] and Wu [Qi] 旁及禪學劍術孫吳兵法擊劍之術 (*pang ji Chan xue, Sun Wu bing fa ji jian zhi shu*) (Zou 1936, vol. 15, p. 15b)". It is safe to say Zou Yi's original text implies that Liu Shu's primary interest was in the Confucian classics; Chan Buddhist texts and works on swordsmanship and military strategy were secondary interests, for the Chinese character "*pang*" above means "collaterally" or "secondarily". Zou Yi's description coincides with what we know about the lifestyle of the Ming Confucian literati, many of whom took up reading Chan Buddhist texts as a hobby. Zou Yi's original text presents Liu Shu more as a loyal woman warrior well versed in the Confucian classics and the texts of other traditions. While Zou does mention at the end of his narrative that Liu Shu built a small hut to spend the rest of her life reading Buddhist scriptures, his text does not single out Buddhism as the key factor that led to Liu Shu's heroic deeds. In contrast, Peng Shaosheng's text reads: "She favored Buddhist scriptures most. Her interests extended to books on swordsmanship and military strategy books by Sun [Wu] and Wu [Qi] 好讀天竺書, 旁及劍術孫吳兵法" (*hao du Tian zhu shu, pang ji jian shu Sun Wu bing fa*). Peng's change of wording creates a difference in connotation. In Peng's version, Buddhist scriptures are no longer secondary among Liu Shu's interests—if Buddhist scriptures do not supersede the Confucian classics on Liu Shu's reading list, they are at least as important.

Ostensibly, Peng intentionally articulated Liu Shu's interest in Buddhism and the importance of Buddhist scriptures in her life even during her early period. In doing so, a link between Buddhism and her later loyal and heroic behavior is established. That is to say, Liu Shu's strong sense of commitment to the state was the result of both a Confucian education and an interest in Buddhism in her youth. It is also obvious that Peng changed Zou Yi's more specific term "*Chan xue* 禪學" (Chan literature) to a more general term "*Tian zhu shu* 天竺書" (Buddhist texts). As mentioned above, Zou Yi's reference reflects the Chan craze among the literati in late Ming. Peng's replacement of Chan with a more general term for Buddhism might well have been out of his prioritization of Pure Land practice over Chan (Wu 2013, pp. 40–42).

Peng's incorporation of the story from a non-Buddhist source into his collection of biographies of Buddhist exemplary women conveys a clear message: Buddhists also embody the virtues of loyalty and filial piety (Peng 1983, p. 151b). Peng's inclusion of Liu Shu's story and sentence manipulation in *Shan nüren zhuan* explicitly suggest that loyalty and filial piety are not only "Confucian" but also "Buddhist" values. Directly following the manipulated sentence, Liu Shu's determination to fight against the rebels and to give up her personal property for this purpose proves that Buddhists never lacked social concern



or ignored societal and familial responsibilities as good subjects and sons and daughters. Indeed, Buddhism helps them to fulfil these social and familial roles all the more.

Another point worth noting is Liu Shu's crossing of gender boundaries to fight in the battlefield. In Peng's version, Liu Shu's trespass is justified by her noble intention to save the country. His version also articulates that it is Buddhism that brought her back to the domestic sphere expected of a widow in Confucian society. Through the cloistered devoted Buddhist life Liu Shu led, Peng argues that, rather than destabilizing Confucian social order, Buddhism reinforced it. Liu Shu, the exemplary Buddhist laywoman in Peng's version, embodied filial piety, loyalty, and chastity—virtues valued by both Confucians and Buddhists.

In the somewhat formulaic narrative of Liu Shu's life by her male biographers, however, the voice of Liu Shu is almost absent. Fortunately, Liu Shu's surviving writing fills the gaps left by her male biographers. These poems voice her loyalty to the Ming dynasty, her anger and pain at the inevitable collapse of the Ming empire, and her final acceptance and transcendence of the harsh reality. Although she might not have intended to fight against male-centered social norms, some of her writings in effect deviate from if not challenge them. Liu Shu's life was restrained; at the same time, she was also empowered by Buddhism and Confucianism. According to Liu Shu's writings, we know that her Buddhist and Confucian identities intersected throughout most of her life. For the convenience of analysis, this paper will discuss how Liu Shu related herself to the two religious traditions in two different sections.

## 3. Political and Military Activities vs. Confucian Gender Norms

According to Confucian gender norms, women's lives were bound by the domestic sphere. Women's roles as daughters (or daughters-in-law), wives, and mothers required them to stay at home to serve their parents with devotion before their marriage and then their parents-in-law after they were married, to be loyal to their husbands whether they were alive or deceased, and to take care of their children. Women were expected to be capable household managers, dutiful wives, and loving mothers. They were tasked with admonishing their husbands and sons to make sure they did not go morally astray. In imperial China, women were not expected to hold any office in the government or participate in any activities in the public sphere designated to men. Instead of direct engagement in public affairs, women served the state by serving the men in their family and helping them to be righteous officials and loyal subjects. In widowhood, a woman was expected to live a secluded life to remain loyal to her deceased husband and maintain her chastity, take care of their surviving parents-in-law, and bring up their children alone. The separation of gender spheres and the distinction of gender roles laid the foundation for the Confucian society.

In her first years of widowhood, Liu Shu did not live a life that a widow was supposed to lead in Confucian society. She not only stepped outside of the inner quarter but entered the domain of men through her expression of political concerns and engagement in military activities. Through Liu Shu's poems collected in the *Geshan ji* 個山集 (Geshan Anthology), this research will explore how a woman's agency and subjectivity were produced not only in what she did but also how she justified her seemingly transgressive behaviors and what she needed to do to negotiate with the gender norms of society. This research is based on the 1992 reprint of the 1914 edition of *Geshan ji*, annotated by Wang Siyuan 王泗原. The collection includes more than nine hundred poems, forty lyrics, and fourteen prose. The 1914 edition was based on three manuscripts: two hand-copied manuscripts were kept by the descendants of the Liu family in Sanshe 三舍 (Liu Shu's hometown), and another one in better condition was preserved by a member of the Liu clan living in Xiangxiang 湘鄉 ([Wang 1992](#)). Although it is difficult to be certain of the authenticity of all the poems collected in the reprinted version of *Geshan ji*, as Wai-yee Li points out "the circumstantial detains and contradictory emotions of these writings convey a sense of nu-

ance and complexity (rather than the ideological purpose one may expect from a forgery) (Li 2010, p. 190)".

In the preface she wrote herself, she stated the purpose of her writing: "[I write] to express my lament of Qishi (Liu 1992, p. 353)". Liu Shu refers to the story of Qishi nü 漆室女 (Woman of Qishi) in the *Lie nü zhuan* 列女傳 (Biographies of Exemplary Women) by Liu Xiang 劉向 (77–6 BCE), a Confucian scholar official in the Han dynasty. The unnamed woman in this story was from Qishi, a city of the Lu state. One day, she leaned against a pillar and lamented the fate of the country. A woman in the neighborhood told her that it was the duty of male officials, not women's business, to worry about national affairs. The woman of Qishi answered, "When misfortune strikes the state of Lu, ruler and subject, father and son will suffer humiliation, and the disaster will spread to affect ordinary people as well. How is it that women alone should escape? I am deeply concerned. How then can you say that women have nothing to do with this?" (Kinney 2014, p. 61) The woman of Qishi was extolled for her wisdom by Liu Xiang and other male Confucian scholars and was held up as an example for women to follow. The woman of Qishi's comment asserted that the fate of a country impacted everyone regardless of one's gender; as such, it provides justification for women to engage in national affairs. Liu Shu employed her allusion to the woman of Qishi to argue that serving the ruler and concern for the welfare of the country transcended gender boundaries. The story of the woman of Qishi thus supports Liu Shu's involvement in political activities outside of the domestic sphere.

Liu Shu's works, especially those poems and essays written to or about her family and friends, suggest how she viewed her gender role as a daughter or daughter-in-law, mother, and widow. These works also reveal how she negotiated the conflicting demands of fulfilling her gender roles and the responsibilities as the subject of the fallen Ming dynasty. Moreover, they also reflect how all of this related to her involvement in military activities.

In addition to the reference to the woman of Qishi, Liu Shu connected her participation in military activities with filial piety. Filial piety is not a gendered virtue and can be demonstrated in many ways. Taking care of one's parents with respect, being obedient to one's parents, and performing proper rituals to honor one's deceased ancestors are all demonstrations of filial piety. Confucius said, "When your father is alive, observe his ideal. When your father passes away, follow his deeds. It is filial piety if you keep your father's way for three years after his death (*Analects*: 1:11)". In other words, to follow one's father's steps is also filial piety. In her eulogy for her father, Liu Shu said, "When I thought of how my father was so devoted to the country and how his determination to fight against the barbarians never died, I sold my jewelry to reward the righteous army. In so doing, I want to let the righteous officials and gentlemen in the country know my father's will" (Liu 1992, p. 366). Liu Shu interpreted her funding and interaction with the military army as a means to carry on her father's moral legacy and to fulfil his unfulfilled aspiration. Thus, her behavior was completely compatible with her gender role as a daughter.

On the other hand, the requirements of filial piety are not always in line with her political and military activities. Some of Liu Shu's poems also indicate the tension between serving her mother-in-law and serving her country. In her poem "Ku gu Hu an ren" 哭姑胡安人 (Crying over the Death of My Mother-in-Law, Madam Hu), Liu Shu writes:

> I grieve the death of my mother-in-law,
> And feel sorry for the mistake of the elder brother-in-law.
> I lived in a remote area for a long time to hide myself.
> How can I return home after experiencing ordeals?
> It is like to going to the end of the world to realize the Way of a subject.
> My devotion to my mother-in-law was separated by the nether world.
> How can I bear to be in sorrow alone,
> When leaning against the wooden door at sunset? (Liu 1992, p. 219)

竟痛安人逝
惟哀伯氏非
僻居久藏跡
經業何其歸
臣道天涯遠
兒情地穴達
何堪獨惆悵
落日倚荊扉

This poem was written upon the death of her mother-in-law when Liu Shu had to flee
from the Manchu conquerors after her aborted military action. It is obvious that she felt
very sorry for having been unable to serve her mother-in-law and to perform the proper
ritual as a daughter-in-law after her death. Her poem indicates the moral dilemma she
had to face: to stay at home to serve her mother-in-law as a filial daughter-in-law or to
fight outside of the domestic sphere as a loyal subject. Facing the inherent contradiction
of realizing the two virtues, she chose loyalty to the state and the ruler. Filial piety to the
elder of the family is considered the foundation of the loyalty to one's ruler, but it is also
praiseworthy if a person puts the ruler and state above family. In addition, it is considered
filial piety if a person can bring honor to their home through serving the ruler and the
state. Liu Shu expresses such kind of sentiment in another of her poems, "Tong ku" 痛哭
(Sorrowful Cry), composed upon the death of her husband. Though stated in an implicit
way, it is safe to assume that her husband died as a martyr in the anti-Manchu war.

> My heart was broken before I could say anything,
> The sound of the wind brought chills to me.
> I feel like walking on frost,
> but I don't care for the remaining ice [on my way].
> Your spirit carries on the moral integrity of a dutiful subject.
> Your shedding blood splashed on the saddle of a hero.
> The Wang clan was fortunate to have an unyielding son like you,
> My tears turn into happiness [when thinking of this]. (Liu 1992, p. 219)
> 未說心先脆
> 聞風膽自寒
> 自知履霜急
> 不信跰冰殘
> 魂續勞臣節
> 血凝志士鞍
> 王門幸不屈
> 哭罷反成歡

The two poems unequivocally prioritize enacting the virtue of loyalty by serving the
country or dying a martyr's death for one's country. Thus, Liu Shu justified her failure
to fulfil the duty of a daughter-in-law through performing greater good. We can also see
her effort to reconcile the conflicting demands of socially accepted gender norms, familial
duties, and devotion to the country by defining filial piety as honoring the family name
through loyalty and moral integrity.

Not surprisingly, Liu Shu encountered disapproval from a family friend for her partic-
ipation in military activities. Her poem, "Song mou gong cong jun" 送某公從軍 (Sending
off an Anonymous Gentleman to Serve in the Army), shows how she responded to the
opposition by upholding virtues such as heroism, loyalty, and filial piety.

> You disapproved of me when I sharpened my sword [to go to the battlefield],
> but you decided to go to the battlefield when I withdrew [from the battlefield].
> I asked you to hide your sword and wait for the right time,
> but your determination is as strong as iron.
> A great man does not seek for personal success,

but to cut through the sun with a sword in hand.
Who does not want to serve the country and repay the love of parents?
 ⋯
In the past, I invited you to die for the country together.
Yesterday, I asked you to wait till the right time.
Today, I wish you will march forward
to wipe off all the big foxes and demonic ants.
You have your aged mother [to take care of] and I have my mine too;
You have a young child, and I have mine.
When you cannot take care of your family for the sake of the country,
I will repay your longtime friendship on behalf of my parents.
Being loyal and righteous to our families and our country,
we will never ignore our families for the sake of loyalty and righteousness.
It is hard to take care of parents and serve the country at the same time.
Send messages to [your family] to let them know you are safe when you serve in the army.
We cannot give up our aspiration to establish ourselves through serving the country,
[This is why you must] drive your horse forward without obstruction.
Those who scoop up water and pick flowers cannot comment on [your decision].
[As] I had swallowed the sword and bit the iron as you will,
[I can understand] the hardship and beauty of your message.
Your noble and lonely ideals can only be represented by the bright moon.
I will bring up your child for you,
So today, you can leave without worry and I wish you good luck.
From now on, in your journey full of wind and rain,
You can only shed tears for your family silently [while you are] hundreds of miles away.
 ⋯
[I hope you will] seize the city to let our flag fly there.
Return home with monumental glory
and serve your aged mother. ([Liu 1992](), pp. 330–31)
我昔磨刀公不悅
而今我隱公偏出
勸公藏刀且待時
其奈公心堅於鐵
丈夫不在取封侯
袖裏青虹宜貫日
報國酬親孰不甘
 ⋯
昔日邀公早殉國
昨日劝公且待時
今日囑公奮前往
封狐赤螳掃無遺
公有北堂余有萱
公有幼兒我有稺
公既為國不有家
余擬為親酬世契
在家在國總忠義
忠義決不將親棄
奉匜絕裾難兩全
馬上勤寄平安字
立身許國奚可辭
還當策馬長趨去

掬水拈花那可評
吞刀嚼鐵余曾試
此中消息美風景
惟將明月鈎孤志
我當為公撫幼兒
公今此去宜得意
風雨蕭蕭已載途
百里暗揮思鄉淚
⋯
奪取孤城立漢幟
銘勳竹帛倏然歸
孺慕長遂椿闈侍

This poem shows how Liu Shu pushed the envelope of gender boundaries through her engagement in politics and military activities. At the same time, she carefully aligned her behavior with the male-dominant ideology. When Liu Shu wrote to a longtime family friend who had disapproved of her military adventure, she resorts to gender-neutral virtues such as loyalty and filial piety and aligns her stance with the views of this anonymous gentleman by articulating the duty a subject owes to his/her ruler and the debt one owes to his/her parents. The way to realize these gender-neutral virtues is gendered. In the time of peace, a woman would fulfil her responsibility as a subject by assisting her husband and son to be righteous and dutiful subjects. In the time of chaos, a woman would maintain her chastity even at the cost of her life as her husband and son died martyrs' deaths for the sake of their ruler. In Liu Shu's case, she did not serve the country through the mediation of one of her male family members (her husband was dead, and her son was still young) and her venture into the male domain apparently deviated from Confucian gender norms. In Liu Shu's poem, she downplays gender distinctions in enacting these virtues; rather, she focuses on commonalities she shares with the anonymous gentleman. She puts herself and the gentleman in the same category of those who "swallowed the sword and bit iron", and she contrasts those who risked their lives on the battlefield with those who "scoop up water and pick flowers", namely those who stayed away from the battlefield and lived an easy life. Usually, "swallowing the sword and biting the iron" is associated with masculinity and "scooping water and picking flowers" with femininity. Liu Shu employs the metaphor for those who fought on the battlefield as applicable regardless of one's gender. In this respect, Liu Shu somewhat changes the way of understanding masculine and feminine roles: masculinity and femininity are related to what one does, not to one's physical body.

In the same poem, she also talks about a shared dilemma: taking care of one's parents and children or serving the country by fighting on the battlefield. In light of this, she offers to "bring up your child for you". Liu Shu here means she would take the place of the gentleman to bring up his child in his stead, and probably would play the role of a fatherly figure instead of a surrogate mother.

In another poem written by Liu Shu, we can also find that she redefines masculinity and femininity by emphasizing one's deeds instead of one's physical features. In this poem to memorize a female friend, who had died a martyr's death to defend her integrity, she writes:

> During the three years, tears and blood has dried like the Xiang River.
> Bitter wind and cold moon cover my small hut.
> Your fragrant spirit is departing the world with a smile.
> Heroines in the boudoir are Great Men. (Liu 1992, p. 264)
> 三載湘江血淚枯
> 苦風寒月掩吾廬
> 香魂欲逝猶含笑
> 閨裏英雄亦丈夫

The Great Man 大丈夫 (*Da zhang fu*) is usually associated with masculinity. It refers to men who were courageous and righteous and behaved in heroic ways. Liu Shu redefines the connotation of *Da zhang fu* and expands the term to refer to a heroic, courageous, and righteous person regardless of the person's gender. In Liu Shu's eyes, a woman martyr who was courageous enough to resist the Manchu invaders—either through taking arms to confront the enemy on the battlefield or by committing suicide in the boudoir to reject the rule of the Manchus—is heroic and could be considered *Da zhang fu*. As such, Liu Shu redefines masculinity as the qualities of heroism, moral integrity, and righteousness and understands it as transcending the gender dualism of man/woman and the gendered spheres of public/domestic.

In fact, in addition to fighting on the battlefield like men, Liu Shu also acted as the head of the household of her natal family—a role that was expected to be played by her older brother or nephew. Liu Shu anthologized and circulated her father's literary works and performed the funeral to bury her father (Liu 1992, pp. 353–57). It was unusual for a married daughter to shoulder the responsibilities expected to be performed by male family members. One possible reason for her having taken over the leadership of her natal family was that her older brother had died, and her nephew had only just come of age (about nineteen years old). She was like the surrogate man of her family (Ko 1992, p. 483). Another reason could be the moral authority she had accumulated as a woman warrior who fought for the country and as a chaste widow who defended her moral integrity with her life. Liu Shu's engagement in military action, in fact, was respected and appreciated by her relatives.[1]

The chaotic transition of the dynasties provided Liu Shu an opportunity to cross gender boundaries to enter the public domain and take up the role normally played by men. Her deviation from the Confucian gender norms did not intentionally challenge the patriarchal system based on the five cardinal relationships defined by Confucianism. Her unconventional behavior was justified by her intention to sustain the Confucian virtues of filial piety, loyalty, and chastity that support the patriarchal system, which set the very gender boundaries that Liu Shu crossed.

## 4. Buddhist Belief and Practice vs. Confucian Gender Norms

According to Liu Shu's biography by Peng Shaosheng, she returned to her hometown and built a small dwelling called Lotus Ferry, where she devoted herself to Buddhist practices for the rest of her life. Unlike her earlier involvement in political and military activities, her return to the domestic sphere in her later years seems to fit Confucian gender norms. In the account of this male biographer, Buddhism domesticated Liu Shu to keep her at home. However, as her poems show, Buddhism not only brought her back to the domestic life that Confucian gender norms expected of a woman, but it also complicated her relationship with the patriarchal system and helped her construct a meaningful social and religious life after the chaos of dynastic transition. Her poems demonstrate changes in her authorial stance and self-positioning, transitioning from a heroic, ambitious, and loyal woman warrior to a Buddhist laywoman who distanced herself from the new dynasty and sought solace, friendship, and spiritual transcendence through Buddhist teachings.

Her poems relating to Buddhist themes reveal her voice and describe experiences in different phases of her life, in which Buddhism constituted an integral part. The two poems that follow, in which she recollects her pilgrimage to *Zhusheng ge* 注生閣 (Zhusheng pavilion) with her husband, exemplify this.

The blue river runs through the purple jade grove,
A thread of incense from my heart reached the blue sky.
The contemplative bodhisattva is so compassionate,
Generously grants sons to [families] to make an amazing addition to the world.

(Liu 1992, p. 244)

碧卷誰開紫玉叢
心香一縷達蒼穹
冥心佛子多情甚
遍贈麟兒補化工

[The patriarch] faced the wall in seated meditation and the brush tip painted
true nature,
Mahākāśyapa, who smiled when the Buddha held a picked a flower, obtained
enlightenment.
The hand that picked the flower tied the knot for
thousands of couples in the world. (Liu 1992, p. 244)
面壁毫端繪本然
拈花一笑四禪天
花中拈就同心結
拋向人間億萬緣

　　The Zhusheng Pavilion, according to Liu Shu's note, was a Buddhist pilgrimage site.
People went there to pray for their sons, and Liu Shu herself gave birth to a son one year
after her trip there. The purple jade grove in this poem refers to grove of purple bam-
boo, always associated with Guanyin Bodhisattva, who was believed to grant sons in the
Chinese Buddhist tradition. Liu Shi expressed her gratitude toward the compassionate
bodhisattva Guanyin who answered her prayer for a son. In the second poem, she refers
to several Chan *gong'an*s. In the first line, "faced the wall" may refer to Bodhidharma, the
first patriarch of Chan tradition, who faced a wall in seated meditation for nine years. The
brush tip comes from Li Tongxuan's commentary on the *Garland Sutra*. It states "self/others
cannot be separated even by a brush tip" to imply the interdependence of all things in the
world. The interdependence theory in the *Garland Sutra* also provides doctrinal ground for
Chan tradition. The second line is related to a famous Chan story, which says that during
a sermon, the Buddha held a flower in their hand without saying anything. The audience
did not understand what the Buddha was teaching by doing so—except Mahākāśyapa,
who responded to the Buddha with a smile. Then, Buddha declared that Mahākāśyapa
truly understood his teaching. In the Chan tradition, this story is often employed to affirm
that the Buddhist truth and experience of enlightenment can be transmitted from mind to
mind without a word. Interestingly, in the last two lines of the poem, Liu Shu creatively
shifted the enlightened minds in Chan tradition to a more secular concern. In her version,
the Buddha and the enlightened patriarchs not only transmit Buddhist dharma but also
bring happy marriages to thousands of people. In these two poems, Buddhism is associ-
ated with the memory of Liu Shi's short but happy marriage. They are more like grateful
expressions of Liu Shu to the Buddha and bodhisattvas who granted her a happy marriage
and the son she wished for. Buddhism, as the two poems indicate, was not only for spiri-
tual transcendence; it also was oriented toward this world and was life-affirming. In other
words, Buddhism, as expressed in these two poems, helped Liu Shu to fulfil the gender
roles expected by Confucian social norms: to get married and to produce a son for her
husband.

　　Liu Shu's happy married life was interrupted by the violent political turmoil of the
Ming–Qing transition. As she was involved in political activism, Buddhism was also closely
related to her political life. In a poem to her husband, she writes:

I carved the vow with my blood to reject food of fish and mutton.
I made this vow in front of the Buddha with great determination.
My mind cannot be changed by time.
Whoever judges the slight difference between the character of "lu" and "yu",
[will not mistake my determination]. (Liu 1992, p. 349)
鏤血絕腥羶
堅心向佛誓
日久不可磨

誰決魯魚字

There are two ways to interpret this poem. It can be read as a vow Liu Shu made to keep a vegetarian diet like many pious Buddhists, according to the literal meaning of "*jue xing shan*" 絕腥羶. If this is the case, then this poem shows her determination to abstain from taking life by adopting a vegetarian diet. This, of course, is an expression of religious piety. On the other hand, the Han Chinese used the term "*xingshan*" 腥羶 to refer to barbarians. In Liu Shu' time, this unmistakably refers to the Manchus, who already occupied a large part of the Ming territory. Considering Liu Shu's involvement in the anti-Manchu resistance, it is more probable that what Liu Shu vowed to renounce is the Manchu rulers. This poem implies Liu Shu's political allegiance under the disguise of her Buddhist practice. The Chinese idiom "*lu yu hai shi*" 魯魚亥豕 refers to unintentional mistakes made in writing Chinese characters with slight differences. This line implies even those who cannot tell the slight differences between two Chinese characters will not misunderstand her determination. In this poem, Liu Shu appealed to Buddha to witness her loyalty to the Ming mandate, which suggests that what Liu Shu valued most and considered as sacred was Buddhism and loyalty to the Ming dynasty.

In another poem, Liu Shu also relates Buddhist practice to her determination to fight for the falling Ming dynasty:

> Who will sacrifice their body to serve the Buddha?
> Vowing to die is like practicing meditation.
> I sharpen my sword to appease my grievances.
> In vain I reach out to heaven to repay the benevolence I received. (Liu 1992, p. 212)
> . . .
> 舍身誰侍佛
> 誓死若依嬋
> 抱怨需磨劍
> 酬恩枉觸天

In this poem, Liu Shu views devotion to Buddhism and serving the country as the same thing. In this sense, Liu Shu shared the same view as her male biographer, Peng Shaosheng, who believed one's Buddhist practice could help people fulfil their responsibilities as loyal subjects. The difference between Liu Shu and Peng Shaosheng is that she considered resisting the Manchu rulers to restore the Ming dynasty as identical to her Buddhist practice; in contrast, Peng Shaosheng viewed serving the Manchu rulers as a demonstration of the loyalty by a good subject. Thus, Liu Shu's Buddhist practice symbolized her defiance against the political regime of the Qing dynasty.

One of Liu Shu's twenty-four *tibi shi* 題壁詩 (poems inscribed on the wall) in a Buddhist temple also indicates the relationship between her political stance and the Buddhist life she would like to lead.

> Seeking the path to immortality, is a vain effort.
> Zhang Liang falsely claimed to fast [as a way] to cheat on the Han Emperor.
> The Buddha land is so graceful and serene.
> I came here to escape from the hustle and bustle of the chaotic world. (Liu 1992, p. 289)
> 求仙無路計空勞
> 辟穀子房誑漢高
> 佛地莊嚴甚幽靜
> 我來游此避塵囂

In the Ming–Qing transition, some Ming officials or literati chose to live a Buddhist monastic life to avoid serving in the Qing government or donning Manchu robes and hairstyles (Wang 1992, p. 387). Such a choice could be interpreted either as passive resistance against the Qing conquerors or as an implicit demonstration of loyalty to the Ming dynasty. Although it was not expected of women to take a position in government or to participate in

any public activities, some women chose to identify themselves as "female remnant subjects" (*nu yi min* 女遺民) of the fallen Ming to distance themselves from the Qing (Li 2010, p. 180). This poem implies Liu Shu's intention to be a female remnant subject.

In this poem, Liu Shu alludes to Zhang Liang 張良 (251–189 BCE), one of the top aides of Liu Bang 劉邦 (256/247–195 BCE), the founding emperor of the Han dynasty. After witnessing Liu Bang kill Han Xing, one of his powerful generals, Zhang Liang resigned by claiming he intended to fast to achieve immortality. Zhang Liang's real intention, of course, was to avoid the same misfortune as Han Xing. As the first sentence implies, Zhang Liang did not attain immortality; instead, hoping to regain political prominence after his resignation, he helped Empress Lü. The line "seeking the path to immortality, is a vain effort" does not necessarily imply the inferiority of Daoism. The allusion to Zhang Liang's false claim of Daoist practice to attain political prominence forms a sharp contrast with Liu Shu's resolve to refuse any relationship with the Qing by seeking refuge in Buddhism. Buddhism provided her spiritual serenity and a physical venue to shelter her from the chaotic life in the Ming–Qing transitional period. In Liu Shu's case, her withdrawal to a cloistered life as a Buddhist laywoman, a choice many women would adopt in their widowhood, was seemingly respectful of long-established Confucian gender norms and tolerated by Confucian ideology. Peng Shaosheng's narrative suggests such an understanding. However, this poem also indicates that her reclusive life was not only a return to the female gender role required by Confucian society; her reclusion was also an expression of passive disobedience. In this sense, Buddhist practices would also have helped her to be the loyal subject expected of Confucian ideology.

One of the poems from "Yi ju wenxin chu bo shu di zhi tong he yi shi ci yun" 移居問心處伯叔弟姪同賀以詩次韻 (To Follow the Rhyme of My Uncles, Brothers, and Nephews Who Composed Poems to Celebrate My Moving into Abode of Inquiring Minds), also relates her Buddhist beliefs to her commitment to the fallen Ming dynasty.

> I promised to die for our Lord.
> Then I turned to Buddhist meditation.
> As a survivor, I am ashamed to face the backbone of the country.
> I look at my shadow and raise my head to look at the clouds.
> Bloody tears streams from my eyes, and I suspect [they are caused by] the sun.
> My heart is as bitter as gentian.
> My remaining soul is stuck at the road of Chu.
> Years have been changed. (Liu 1992, p. 212)
> 我許君王死
> 無何居士禪
> 偷生愧鼎石
> 顧影望雲天
> 淚滴丹疑日
> 心癡苦似蓮
> 殘魂羈楚道
> 歲月已頻遷

This poem also reveals that Chan Buddhism provided her a way to withdraw from the world following her failure in restoring the Ming dynasty. In this poem, Liu Shu expresses a sense of guilt as a survivor and the sorrow of a loyal subject after witnessing the fall of her country. It is also clear that she has rejected subordination to the new dynasty and does not want to have anything to do with the new ruler.

Several poems by Liu Shu suggest that Buddhism played an important role in her social life. She did not live a secluded life as her male biographer portrayed; in fact, her social network extended to Buddhist monastics, as evidenced by a number of her poems. In one example, "You ni zi yang shan chi can yu guo fang" 有尼自仰山持參語過訪 (A Nun from Mount Yang Visited Me with Phrases to Contemplate on), she writes:

> Where did the Blue Bird cross the green cloud,

and come to my mountain home despite the wind and rain?
Reciting spells and meditating on the Supreme knowledge,
She has realized the three lives are illusions like bubbles and deliverance is available.
The dewdrops on the thousands of leaves are like pearls dotted on the branches.
A lotus is blooming under her seat, and her body is covered with flowers.
While we were talking about the message leisurely,
we try to scoop up the moonlight [pouring] on the steps. (Liu 1992, p. 300)
何處青鸞度碧霞
滿頭風雨到山家
摩訶萬呪參無等
泡幻三生渡有涯
珠綴梢枝千葉露
蓮開寶座遍身花
坐中且漫談消息
試向階前挹月華

In the first two lines, Liu Shu compares a nun from Mount Yang to the miraculous and auspicious Blue Bird that brings good news to people. The next four lines extol the spiritual achievement of the nun. The last two lines describe lighthearted and joyful discussions of Chan phrases that she had with the nun. As Beata Grant points out, friendship between monastics and the laity is not only found among men; it is also found among women (Grant 2010, p. 246). Buddhist teachings and friendship with Buddhist monastics represent an alternative to the dark and sorrowful reality that Liu Shu found herself in.

Buddhism also connected Liu Shu to women who shared similar life experiences. In her anthology, there are nine poems dedicated to Madam Zou 鄒夫人 (dharma name Huiyin 慧印). Liu Shu addresses her as *chanyou* 禪友 (Chan friend). According to Liu Shu's poems, we know that the two women shared commonalities: both of them were widows; their husbands probably died as martyrs in the resistance against the Manchu invaders or rebels; they raised their children alone; and, more importantly, they shared interests in Chan Buddhism. The following is one of the poems Liu Shu wrote for Huiyin:

The remaining music regrets the broken harp.
The Xiang flute cries over the fall cicada.
The blood of an upright person turns into blue mountain.
Pure heart is enshrined in the white lotus.
[We have] long a discussion of Buddhist texts in the moonlight.
[Our] inquiry of the Buddhist dharma reaches as high as the sky.
[Zhaozhou] washing the bowl surprised the dragon girl.
[We] smile at each other with a deep understanding of the [Buddhist teaching and each other]. (Liu 1992, p. 227)
餘音悲琴斷
湘笛咽秋蟬
碧血凝青嶂
冰心放白蓮
談經長照月
問法遠參天
浣鉢驚龍女
會心一笑傳

This poem describes the stages in the violent Ming–Qing transition that Liu Shu and her friend had both experienced. The first two lines of the poem are about the shared sorrow of the loss of their beloved ones and the collapse of the Ming dynasty. The third and fourth lines praise the moral integrity of Huiyin and probably her husband. The fifth and sixth lines describe their keen interest in Chan and their inquiry into Buddhist doctrines. In the seventh line, Liu Shu employs two Buddhist allusions. The first one is to a Chan *gong'an*.

When a monk asked Zhaozhou how to practice Chan, Zhaozhou replied, "Have you taken you porridge?". After the monk said, "Yes", Zhaozhou replied, "Wash your bowl". The monk was enlightened. The second allusion is to the dragon girl in the *Lotus Sutra*, who obtained enlightenment in an instant after she presented a gem to the Buddha. By crafting an imaginative occasion of the meeting of two enlightened minds, the last two lines imply the shared joy of the two like-minded friends when they obtained a breakthrough in Chan practices and realized they understood each other so well. It is also notable that employing a constructed meeting between an enlightened male Chan master and a girl refers to the shared joy of spiritual breakthrough between two women, indicating that an enlightened mind transcends the boundaries of the male/female binary. The two women found solace and comfort in their shared interest in Chan and the endurance of their friendship in their traumatic lives in the Ming–Qing transition. It also reflects Liu Shu's change in her emotions and aspirations. Her Buddhist practices enabled her to overcome the pain and anger brought about by the collapse of the dynasty and the loss of loved ones. She has accepted reality, found a new way to maintain her moral integrity, and been healed by her religious belief and practice.

The above cited poems indicate not only that Buddhism was deeply integrated into Liu Shu's life but also that it had different connotations at different stages of her life. In the time of peace, Liu Shu's secular and intellectual life was enhanced by Buddhism. Having chosen to live a cloistered life as a Buddhist laywoman, on the one hand it provided her a shelter from the chaos in the outside world. On the other hand, Buddhism symbolized her determination to maintain her moral integrity as a loyal subject of the fallen Ming dynasty. Through Buddhism, she constructed a social network with like-minded women. Buddhist teachings and practices enabled her to enjoy friendship with women who shared similar traumatic experiences; it also helped her transcend the pain and anguish brought about by the loss of loved ones in the violent dynastic transition.

## 5. Conclusions

Liu Shu's life is indicative of the relationship between women's agency and the dominant social structure. She crossed gender boundaries and engaged in political and military activities normally considered to be within men's domain. In this sense, her behavior could be considered transgressions that challenged Confucian gender norms. In another sense, her military and political engagement could be placed in the narrative framework of filial piety and loyalty to the fallen Ming dynasty to justify her "transgressive" behavior. As such, her expressed intention was to maintain the dominant structure represented by the patriarchs of the family and the country. Thus, she was accepted and extolled by male elites of the Ming and Qing dynasties. Yet, her Buddhist practices and her return to the inner quarter in accordance with Confucian gender norms had a defiant connotation against the newly established dynasty that epitomized patriarchal authority. Alternatively, she created a meaningful life amidst a gloomy reality by seeking spiritual transcendence in Buddhist teachings and practices and finding solace and support in her friendship with Buddhist female monastics and laywomen. To summarize, to understand women's agency and the dominant structure, we cannot simply define agency as resistance against the patriarchal system. The patriarchal structure is not monolithic. Rather, it is full of conflicts and contradictions. These contradictions and conflicts, especially in a time of social and political upheaval, allow women to interact with the patriarchal system in different ways. Liu Shu's life demonstrates how a woman could defend, uphold, and reproduce the virtues that supported and enhanced the patriarchy through going beyond gender boundaries and redefining masculinity and femininity to live a life of independence and empowerment.

**Funding:** This research received no external funding.

**Institutional Review Board Statement:** Not applicable.

**Informed Consent Statement:** Not applicable.

**Data Availability Statement:** Not applicable.

**Conflicts of Interest:** The author declares no conflict of interest.

## Notes

[1] Liu Shu wrote a series of twelve poems entitled "Jun shi wei bi jia ren quan wo yi gui" 軍事未畢家人勸我以歸 (The Military action was not over, and my family persuade me to return home). At least two poems reveal the supportive and accepting attitude of her family towards Liu Shu's military adventure.

The sound of the horn rouses my heroic spirit,
Sisters removed my treasured saber from me.
Nothing in the mountain worries me.
Let Pulao [mythical animal] roar in the sea. (Liu 1992, p. 248)
一聲畫角興方豪
姊妹為余脫寶刀
從此山中無個事
凭他橫海吼蒲牢

I am ashamed to compare my service in the army with Mulan.
I mounted on a light saddle again after I returned home.
Reviving my home,
I am happy to reconnect with the elders and old friends. (Liu 1992, p. 248)
此次從征愧木蘭
還鄉又復上輕鞍
依然重整粉榆社
父老猶聯故舊歡

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
