# Peer review of "Sword and Lotus: The Life of a Confucian Buddhist Woman Warrior in Seventeenth Century China"

_religions, doi:10.3390/rel14060739_

Round 1
Reviewer 1 Report
Title: “Sword and Lotus: The Life of a Confucian Buddhist Woman Warrior in the Seventeenth-Century China”
Review recommendation: Acceptance with minor revisions
Reviewer’s comments:
This essay offers a well-researched study of the life and works of a late Ming author Liu Shu (1520-1657) who was loyalist warrior and Buddhist poet. Through a contextualized examination of Liu Shu’s life account, her political engagement in anti-Manchu military uprisings, her reconfigurations of Confucian gender norms, and her devotion to Buddhist practices, the author demonstrates Liu Shu as a representative case of women’s social, political, and spiritual agency beyond the modernist discourse of overt resistance against a patriarchal system. Instead, the author convincingly argues that women authors in the polemical era of Ming Qing transition took to more complicated and engaged activities in writing, military leadership, religious self-cultivation to express new imaginaries about individual autonomy, subjectivity, moral paradigms, as well as ideals of the nation. The essay contains original research on an important research topic at the intersection of religious studies, women and gender studies, premodern women’s engagement with representations of the nation-state. The analysis is well supported through rich scholarship and appropriate citations. In addition, the poems selected from the works of Liu Shu are persuasive examples for the author’s analysis and are well translated. The overall quality of the essay convinces me that it could be accepted for publication with minor revisions. Below are some suggestions for the author to consider as he/she completes the revision process.
Suggestions:
1. On structure of the essay: whereas the overall structure of the essay is lucid and functional, I wonder if the author could create more dialogue between these sections rather than listing them as a topic after another. For example, the introduction section, though interesting in establishing a comparative theoretical framework on how to approach and understand feminine resistance against patriarchy in premodern context, could be further amplified by adding more grounded discussions on current scholarship on women’s political passion and resistance through writing and personal engagement in the Ming and Qing. A little bit more scholarship could be engaged here. In addition to Wai-Yee Li’s seminal monograph, Grace S. Fong, Susan Mann, Hu Siao-chen, Qian Nanxiu and many others have published profusely on how women poets utilized poetry to negotiate or even redraw the boundaries between gendered spheres and modes of expressions, and how Ming Qing women poets took to political issues in writing to inscribe new forms of femininity/ feminine subjectivity as they coped with new forms of subjective experience due to travel, forced exile, chaos and disorder, banditry and uprisings. Check the Brill anthology The Inner Quarters and Beyond for citations. Similar scholarship on women’s political ambition and aspirations could also be found in fiction and drama during this period. Check Maram Epstein’s important article on the tanci fiction Heaven Rains Flowers 天雨花. All of these are on works by Ming Qing women authors that displayed political interests, ambitions, or aspirations. To ensure that this current study creates a meaningful dialogue with published scholarship, a bit more discussion of this in the introduction session will show how the essay could enrich these current studies by adding another important author study to the repertoire (and also bringing in new perspectives in gender and religious identities in Ming Qing transition period).
In addition, section two and section three, as the author foreshadowed, maintain interconnections and should not be read as separate parts. I wonder if some discussions could be added to help with the transition of the two sections. For example, tales of historical militant women in premodern China are many; how does Liu Shu’s tale contribute to our understanding of militant women and their political activism at a time of war and chaos? (Check Louise Edwards’ 2010 article on transformation of women warrior’s image esp. Mulan.) In section three (the strongest section of the essay), I wonder if Liu Shu, who by this time retired from her former militant life, was more interested in fashioning a form of new, gendered religious identity (eg: as the reference of 龍女 and the mutual poetic exchanges with the nuns) than searching for voices of resistance and rebellion. Whereas her later poems are indeed very expressive of grievances about a dark social and political reality (by referring to loss, death, and exile, eg. P17), clearly the poetic voice changed from some earlier poems that were more directly evocative of political themes such as courage, ambition, sacrifice and martyrdom. The recurring references to “會心一笑” in her later poems seems to suggest a spiritual transcendence because of practices of Buddhism rather than a unbending commitment to political resistance. Hence the last paragraph in section 3, on page 15, might need to be amplified and rewritten to address this possible difference or change in authorial stance and self-positioning. --- On a resonant note, I wonder if the author could build on her findings about Ming Loyalist Buddhist women’s poetic exchanges as discussed on page 14, and add a bit contextualization and analysis on how this important group of women authors’ poetic creations expand the horizon of current studies on Ming Qing women’s literary, cultural and social engagement. This is a very important research repertoire and could lead to further discoveries and researches. It will be necessary to discuss how religious aspirations, political visions, and personal experiences intersected in these women’s writings and created new gendered voices.
Also a small point: on page 12, the reference to 腥羶 may not necessarily have a racial or ethnic reference to Manchu traditions. Food abstinence and vegetarianism was practiced among Han Buddhists of the time widely. This could be just a declaration of her determination to take to religious self-cultivation and abstinence through daily practice. Unless there is at least another example from her works that uses this term with a clear political reference, I would recommend removing the interpretation of it as Manchu traditions, and instead focus on her commitment to religious practice. On the same page, 抱怨需磨劍, 酬恩枉觸天. The author translated the lines as: “I sharpen my sword to revenge my feud, / It is like to reach to the sky to repay the benevolence of my lord.” I wonder if the author should retranslate this? If at this time she was converted to Chan Buddhism, 抱怨 (literally “holding grievances”) should not be understood as “revenge” which is deeply disapproved by Buddhist classics. Instead, it could be to “redress injustice”, “to appease my grievances.” The following line is more like “In vain I reach out to heaven to repay the benevolence I received.” Some re-reading of the context of this poem (time, context, situation, audience) will be necessary to do a correct translation of it.
Similarly, page 13, the poem with the line “辟穀子房誑漢高”, I wonder if the author might have embedded a tone of irony intentionally to show her political and religious integrity, not so much against someone who practiced Daoism, but rather, as the author said, some of her male peers of the time who “feign” religious or political integrity through a performance like Zhang Liang but in actuality were still hoping to gain new political prominence. A bit more discussion on this would be helpful (if there was such a context). ---On a related note, because Liu Shu’s case shows a fusion of Confucianism and Buddhism in women’s literary creations and religious practices, a bit discussion on the relations between Confucianism, Buddhism, Daoism as shared influences on premodern Chinese secular religious beliefs will be nice. Perhaps just one or two essays about this in Ming Qing period will help.
Another suggestion is about in-text citation numbers. Whenever alluding to a scholar and his/her works in the text, it is always necessary to give year, page number, and use quotations if directly quoting. For example, page 14, the page number for the quote from Beata Grant should be added. There are quite a few examples of which that can benefit from clearer in-text citations. The author may need to do a self-check on all resources and make sure if they are evoked in the analysis, they each have an in-text citation with year and page numbers.
In light of copyediting, the author may also want to have the revised essay copyedited. There are a number of places when the discussion looks generalized. In the abstract and in the text, “…by the patriarchal ideology” is not right. Perhaps delete “the.” In the title, “in the Seventeenth-Century China” is not right. Delete “the” before “seventeenth.”

Author Response
First, I want to express my deepest gratitude for the reviewer’s thoughtful feedback and insightful suggestions.
The following is my response to the reviewer’s suggestions:
- On the structure of the essay: in the introductory part of the essay, I have added a paragraph to create dialogue with published scholarship. Please see p. 2 (line 56-line 84) according to the suggestion of the reviewer. Thanks for the references.
- On the connection between Section Two and Section Three: I added some discussion to highlight the role of Buddhism in the transition of Liu Shu’s life from a woman warrior to a Buddhist laywoman. See p. 12 (line 568-line 573), p. 16 (line 802-line 803), pp. 16-17 (line 814-817, line 819-line 833).
- I respectfully disagree with the reviewer’s suggestion about the interpretation of the Chinese word xingshan 腥羶 because there are some Ming literary works that use xingshan to refer to the barbarians in the north. For example,
Yao Maoliang 姚茂良 (1465-1489) wrote “率百万之师,决千里之胜,扫荡腥羶,殄灭无遗,庶可以雪国家之耻。I will lead millions of soldiers and win the battle thousands miles away to wipe out xingshan (the barbarian invaders), and exterminate all of them to revenge the national insult. ”
Feng Menglong馮夢龍 (1574-1646) in his Chronicle of the Eastern Zhou “老夫年邁無識,止為臣子,義不容辭,勉力來此。掃蕩腥羶I am so old and ignorant, but it is my righteous responsibility as a subject to exert my utmost effort to wipe out the xingshan (barbarians).”
in another poem, Liu Shu writes, “鏗訇匕首嘯新硎,牧雨千花綴翠屏。洛下春殘無宿火,長虹欲吐掃羶腥。The short dagger is honed on the new whetstone, thousands of flowers bathed in the rain decorate the green screen. No fire remained in the end of the spring, the long rainbow will rise to sweep Shanxing 羶腥” This poem alludes to Jingke 荊軻 (the dagger, and the rainbow), who assassinated the King of Qin. In this context, Shanxing here does not refer to lamb and fish, but to invaders.
- The interpretation of 抱怨需磨劍, 酬恩枉觸天. I have revised the translation according to the suggestion of the reviewer. See p. 14 (line 662-line 663)
- The poem with the line 辟穀子房誑漢高. I have added the context and revised the interpretation of the poem. See p. 14 (line 692-line 710)
- I have revised the in-text citations and deleted “the” in “the patriarchal ideology” and “the” in Seventeenth-Century.
Thank again for the reviewer’s comments and suggestions.
Reviewer 2 Report
This is an excellent paper. The translations are smooth and faithful. Close readings are careful and inspiring. Once published, it will profoundly enrich the ways we teach the history of late imperial China. Below are some suggestions. Most of them are minor issues. However, readers would appreciate it if the author could situate this fascinating case study in the context of recent developments in the field of gender history of China. Also, I wonder if the author could make it explicit upfront how Buddhism mattered in this story. This being said, the author should enjoy the freedom of taking or disregarding any suggestions below. In this sense, I recommend the journal publish it in whichever form the author and the special-issue editors deem fit.
p.1 “[a woman,] as an intelligent and moral agent”
A woman is always “an intelligent and moral agent.” I would not add this phrase here.
p.2 “ventured into male-defined spheres,”
It is not crystally clear to me what these spheres are in this context. Warfare, politics, or religion? Or all of them?
p.2 “ … a woman with subjectivity could interact with the patriarchal system other than intentional resistance against it. On the other hand, patriarchal institutions are not always consistent in their requirements for women’s virtues, especially in a time of dynastic transition and the shift of political allegiance; consequently,”
First, I would delete “with subjectivity.” Second, Dorothy Ko has already made this point in her Teachers of the Inner Chamber (p.9) that a woman could do many things without open resistance against patriarchal system. In addition, the author might need to refer to previous studies focusing on woman warriors/commanders of other periods. See, for instance, Shoufu Yin, “Rewarding Female Commanders in Medieval China: Official Documents, Rhetorical Strategies, and Gender Order.” Journal of Chinese History 6.1 (2022): 23–42. Wu Pei-Yi, “Yang Miaozhen: A Woman Warrior in Thirteenth-century China,” Nan Nü: Men, Women and Gender in China 4.2 (2002): 137–169. In brief, there is a growing body of scholarship on women and warfare in imperial China. Readers would appreciate it if the author could situate this fascinating case study in the context of recent developments in related historiographies.
p.2 “It is hard to tell which version is more historically accurate but examining how these male writers represented Liu Shu’s life was like may help us to understand why she was represented in a certain way and what the agenda these male literati may have been advancing.”
The sentence is a bit clunky. “It is hard to tell which version is more historically accurate, but examining how these male writers represented Liu Shu’s helps us understand the intellectual/ideological agenda of the male literati who detailed her deeds.”
p.2 “To compare Peng’s biographical account of Liu Shu and her own writings enables us to look beyond a male-constructed exemplary Buddhist laywomen embodying Confucian virtues to see how Liu Shu led a life of autonomy through upholding values that reinforced the Confucian patriarchal system.”
A bit clunky. To compare Peng’s biographical account of Liu Shu and her own writings enables us to see how Liu Shu led a life of autonomy through upholding values that reinforced the Confucian patriarchal system.”
p.5 “Fortunately, Liu Shu’s surviving writing fills the gaps left by her male biographers.”
A skeptical reader might raise the (somewhat unfair) question: whether it is possible that others had composed these words in her voice. To forestall these kind of doubts, the author might cite Li Wai-yee 2014, esp. Chapter 3. In any case, the textual history of any writings cited is of pinnacle importance.
p.5 “At times, her voice as revealed through her literary works in line with the patriarchal ideology; at other times, it deviates from the male-centered social norms.”
Lacking a verb in the first part. A bit vague overall. The author might point out that she writes her poems in specific contexts for specific purposes. However, while it is not her plan to build a grandiose theory against the male-centered social norms, some of her writing in effect deviate if not challenge them.
p.6 “Her literary works were anthologized in the Geshan ji 個山集 (Geshan Anthology).”
Readers would want to know more about this collection. Who compiled it? When? Is it printed?
p.9 “This poem shows how Liu Shu pushed the envelope of gender boundaries.”
I am not sure whether it is true, and if so, how. Later, the author writes “she downplayed the gender distinctions in enacting these virtues, but focused on commonalities between her and the anonymous gentleman.” I think this is excellent. I would tighten this long paragraph alongside this point.
p.10 “In the same poem, she also talked about a shared dilemma: to take care of one’s parents and children or to serve the country by fighting on the battlefield, and thus she offered to “bring up your child for you.” Apparently, Liu Shu here means she would take the place of the gentleman to bring up the child. She would act more like a fatherly figure instead of a surrogate mother.”
Is this point necessary? I am not sure. I cannot tell whether she would act like a fatherly figure ...
p.12 “Whoever judges the slight difference between the character of “lu “and “yu.””
What is at stake here?
p.15 “In Liu Shu’s life, Buddhism not only brought her back to the domestic sphere to reassume her gender role, but also provided symbols and alternatives to reject the Manchu rulers.”
This is very helpful. Readers would appreciate it if the author could make it clear upfront how Buddhism mattered. As the author shows, Buddhism, in this case, is not liberating women from Confucian patriarchy (p.12) but does provide a space to “reject the Manchu rulers.” Overall, this paper would be even stronger if the author could explicitly address this question: How Buddhism matters?
p.15 “her transgressive behavior was motivated by her filial piety to her father and loyalty to the ruler of Ming Dynasty, so her intention was to maintain the dominant structure represented by the patriarchs of the family and the country.”
Are we sure that her behavior was motivated by her filial piety? Or rather, putting her military achievements in the narrative/conceptual framework of filial piety helps elites of her times to accept her “transgressive” behavior?
Author Response
First, I want to express my deepest gratitude for the reviewer’s thoughtful feedback, and very detailed and insightful suggestions.
- 1. I deleted “an intelligent and moral agent” according to the reviewer’s suggestion. See p. 1 (line 33).
- 1. I specified “male-defined spheres” as “warfare and politics.” See p. 1 (line 34).
- 2. I deleted “with subjectivity.” See p. 2 (line 50). I also added a discussion of recent scholarship on women and warfare in imperial China. See p. 2 (line 56-line 84).
- 2. I changed the sentence to “It is hard to tell which version is more historically accurate, but examining how these male writers represented Liu Shu’s life helps us understand the intellectual/ideological agenda of the male literati who detailed her deeds.” See pp. 2-3 (line 97-line 100).
- 2. I changed the sentence to “To compare Peng’s biographical account of Liu Shu and her own writings enables us to see how Liu Shu led a life of autonomy through upholding values that reinforced the Confucian patriarchal system.” See p. 3 (line 105-line107).
- 5 and p. 6. Regarding the authenticity of Liu Shu’s writing and her anthology Geshan ji. I cited Wai-yee Li’s statement on the authenticity of her writing, and also added some background information of her anthology. See pp. 6-7 (line 305-line 315).
- 5. I changed the sentences about the voices in Liu Shu’s poems according to the reviewer’s suggestion. p. 6 (line 274-line 278).
- 9. “This poem shows how Liu Shu pushed the envelope of gender boundaries.”
I reframed this sentence and added “through engagement in politics and military activities” to explain how she pushed the envelope. See p. 10 (line 489-line 491).
- 10. Regarding the verse “I will bring up your child for you,” I think it implies that she would take the place of this gentleman to bring up the child when he was absent or was killed in the battlefield, so it is probable that she would act like a fatherly figure instead of a surrogate mother. See p. 11 (line 515-line 517).
- 12. “the difference between the character of ‘lu’ and ‘yu’”. I have added an explanation to the allusion of “lu” and “yu.” See p. 13 (line 651-line 654).
- 15. I have added some sentences to tell the readers how Buddhism matters in Liu Shu’s life. See p. 2 (line 82-line 84), p.12 (line 568-line 573).
- 15. I have changed the statement according to the reviewer’s suggestion. See p. 17 (line 839-line 848).
Thanks again for the reviewer’s comments and suggestions.